# URECA: UNIQUE REGION CAPTION ANYTHING

## ABSTRACT

Region captioning models often struggle to generate descriptions *unique* to a specific area of interest, instead producing generic labels that could also apply to other regions within the same image. This ambiguity limits their effectiveness in downstream applications and prevents them from capturing the fine-grained details that distinguish objects. To address this, we introduce the **Unique Region Caption Anything (URECA)** dataset, a new large-scale benchmark designed to enforce caption uniqueness for multi-granularity regions. URECA dataset is constructed using a novel four-stage automated data pipeline that establishes a one-to-one mapping between a region and a descriptive caption, ensuring that each description uniquely identifies its target. We also propose the URECA model, an architecture built on two innovations for generating unique region captions: a decoupled processing strategy that preserves global context by separating region and image inputs, and dynamic mask modeling to capture fine-grained details regardless of any input image scale. Code and weights will be publicly released.

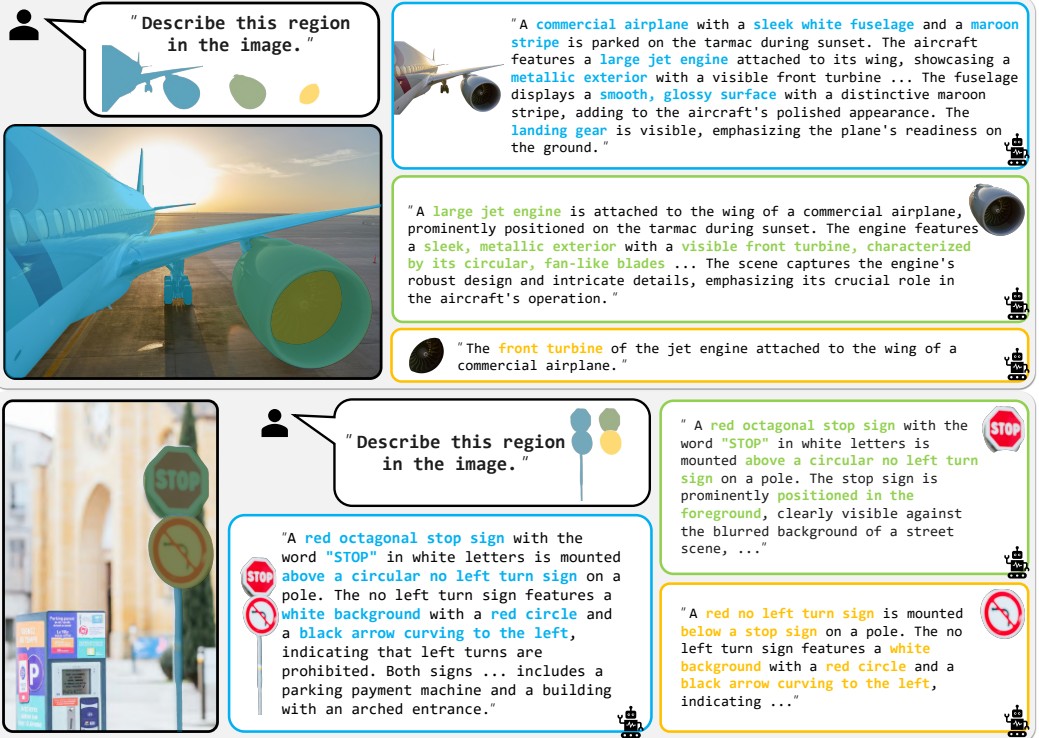

Figure 1: We introduce the **Unique Region Caption Anything (URECA)** dataset, a novel region-level captioning benchmark designed to ensure caption uniqueness and support multi-granularity regions. Each caption is uniquely mapped to its corresponding region, capturing distinctive attributes that differentiate it from surrounding areas. Furthermore, we propose the **URECA** model, trained on our dataset, which effectively generates unique captions for regions at any level of granularity.

# 1 INTRODUCTION

Image captioning is a long-standing task at the intersection of computer vision and natural language processing, requiring both a deep understanding of visual content and the ability to generate fluent descriptive text. Success hinges on the synergy between accurate visual perception and coherent language generation. Building upon this foundation, region captioning presents a more granular and complex challenge, where even state-of-the-art Vision-Language Models (VLMs) (ModelScope, 2024; Wang et al., 2025; OpenAI, 2024; Team et al., 2023) face significant open problems. Unlike image captioning, which provides a holistic but often shallow description of an entire scene, region captioning (Yu et al., 2016; Krishna et al., 2016; Sun et al., 2024; Yuan et al., 2024; Wu et al., 2022; Fanelli et al., 2024; Lai et al., 2024) demands that a model describe all relevant elements within a specific, user-defined area of interest.

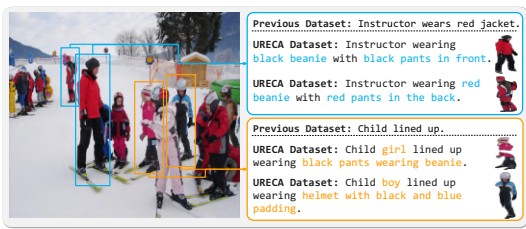

Prevailing research in region captioning has largely focused on improving the fidelity of descriptions by using precise localization inputs, such as 2D coordinates, bounding boxes (Huang et al., 2024; Wang et al., 2023; Wu et al., 2022; Zhao et al., 2025), and masks (Rasheed et al., 2024), to capture fine-grained details. While these methods have achieved impressive results in generating detailed text, they often overlook a critical requirement: caption **uniqueness** within a single image. We formally define a caption as unique if it only refers to its designated region unambiguously within the context of the image, such that the description cannot be correctly applied to any other region in the same image. For example, as illustrated in Figure 2, distinct regions containing different instances of the same object class (e.g., two different women in an image) may be assigned identical, generic captions.

Figure 2: **Limitations of existing datasets.** This figure shows examples of region–caption pairs from widely used datasets (Krishna et al., 2016), where a single caption often applies semantically to multiple bounding boxes, resulting in a one-to-many mapping. In contrast, our URECA dataset provides unique captions for regions.

This failure to generate unique descriptions introduces significant ambiguity. It can cause errors in downstream applications like referring segmentation (Ding et al., 2025), which relies on a description to uniquely identify a target object. Moreover, it can confuse the model during training, as it is forced to map visually distinct inputs to identical ground-truth captions. We identify three key obstacles hindering progress:

1. **Lack of uniqueness-driven datasets.** Existing datasets (Krishna et al., 2016; Yu et al., 2016; Rasheed et al., 2024; Zhou et al., 2024) are not explicitly designed to enforce a one-to-one mapping between a region and its description. Their captions are often generic and can be reused across different instances of the same object class, thus failing to capture distinguishing visual characteristics.

2. **Poor granularity in annotations.** High-quality annotations are scarce, especially for non-salient or complex regions. Many datasets focus only on prominent objects, neglecting parts of objects, object-to-object relationships, and background elements that are crucial for comprehensive and unique descriptions.

3. **Lossy region encoding.** Despite their strong generative capabilities, many VLMs (Chen et al., 2023; Heo et al., 2025; Lian et al., 2025) process regional inputs in a lossy manner. Their architectures often downsample or simplify region masks, discarding the fine-grained spatial details crucial for distinguishing between similar instances. This problem is particularly severe for multi-granularity regions (e.g., small objects, thin parts), fundamentally limiting the model's ability to perceive the visual cues required for a unique caption.

To address these fundamental challenges, we introduce the **Unique Region Caption Anything (URECA)** dataset. URECA dataset is large-scale resource specifically designed to provide unique captions for multi-granularity regions, a contribution largely absent from previous research. To achieve this, we developed a meticulous four-stage data pipeline that enforces a one-to-one mapping between textual descriptions and their corresponding visual areas. Unlike existing datasets that are

often limited to salient objects and generic phrases, URECA dataset encompasses a diverse range of subjects including objects, parts, and backgrounds, ensuring that every caption uniquely identifies its region.

To properly evaluate a model's ability to generate captions that are both unique and accurate, we created a specialized test set with an additional verification stage to ensure data quality. Furthermore, we challenge the reliance on traditional metrics (e.g., BLEU (Papineni et al., 2002), CIDEr (Vedantam et al., 2015)), arguing that for uniqueness, semantic equivalence is more critical than the exact lexical overlap they reward. We therefore demonstrate that LLM-based evaluation metrics (Lian et al., 2025) can effectively assess semantic quality while maintaining a high correlation with traditional scores.

To leverage the fine-grained knowledge within our dataset, we propose the **URECA model**. Its architecture is founded on two key technical innovations. First, we introduce a decoupled processing strategy, where a dedicated mask encoder processes the region prompt into spatial tokens while the full image features remain unaltered. This preserves the global context by avoiding destructive modifications to the input and precisely locate the prompt region. Second, to handle these region prompts with high fidelity across all scales, we employ a dynamic mask modeling technique that systematically tiles the mask, overcoming the fixed-input resolution limitations of visual encoders.

Experiments validate the effectiveness of our approach, demonstrating that URECA successfully interprets region prompts to generate detailed, unique captions that are precisely grounded in the target area. We believe that our model, dataset, and insights will significantly advance research in this domain and broadly benefit the vision–language community.

## 2 RELATED WORK

Although MLLMs have demonstrated impressive image understanding capabilities, generating captions for specified regions remains a challenging task. LLaVA (Liu et al., 2023) and MiniGPT-2 (Chen et al., 2023) have explored conditioning on regions by translating bounding box coordinates into natural language tokens. However, this approach relies heavily on the MLLM's abstract ability to map textual coordinates to spatial locations. Other methods attempt to overlay region masks or contours directly onto the image (Cai et al., 2024; Yang et al., 2023c; Shtedritski et al., 2023). While intuitive, this permanently alters the original image, obscuring valuable contextual information.

To avoid modifying the image, another line of work performs feature pooling directly from the vision backbone's feature maps, conditioned on either bounding boxes (Wu et al., 2022; Dwibedi et al., 2025; Ma et al., 2024; Zhang et al., 2024a) or masks (Guo et al., 2024; Heo et al., 2025). While masks provide more precise localization than ambiguous bounding boxes, the pooling operation itself introduces significant drawbacks. It is typically performed on low-resolution feature maps and aggressively aggregates spatial information, leading to a loss of fine-grained details such as shape and boundaries. In extreme cases, features for small regions can vanish entirely. Critically, all feature pooling methods share a more fundamental limitation: by extracting features only from the target region, they discard the surrounding global context. This loss of contextual information makes it nearly impossible for the model to capture the distinctiveness required for unique caption.

In summary, prior works have not adequately addressed the challenge of generating captions that are simultaneously unique, precisely localized, and applicable across multiple granularities. This gap stems from two primary factors: the lack of a large-scale dataset designed to enforce caption uniqueness, and architectures that struggle to process regional prompts without sacrificing either fine-grained detail or global context. To bridge this gap, we first propose an automated pipeline for generating a multi-granularity dataset with unique captions. We then present a novel model architecture specifically designed to leverage this data, which preserves both local attributes and global relationships, enabling it to generate truly distinctive descriptions.

## 3 URECA DATASET

In order to generate unique caption from VLMs, high quality dataset with unique caption pair with region are crucial. To this end, we propose URECA dataset pipeline, that made with four-stage approach, enabling to build a large and diverse granularity levels with high quality unique captions.

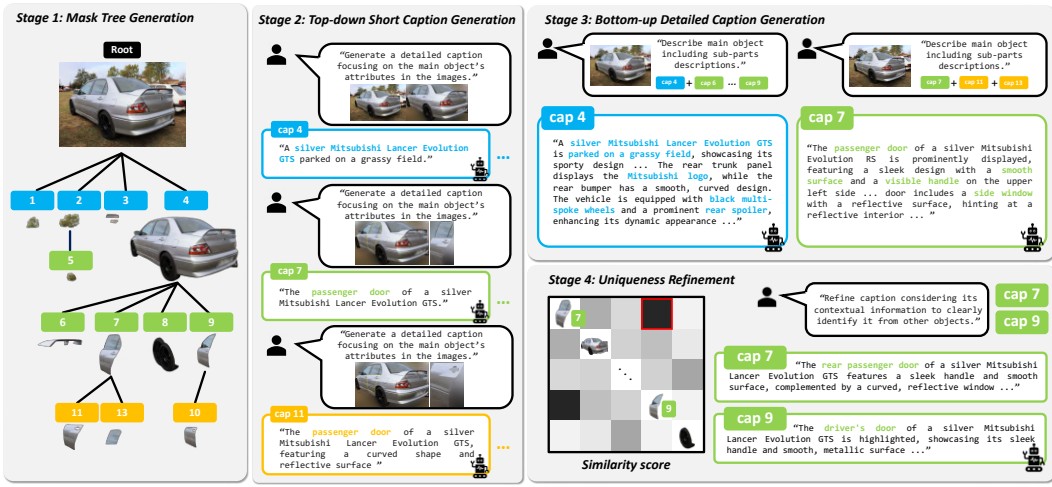

Figure 3: **Automated data curation pipeline of URECA dataset.** Our pipeline consists of four key stages to generate unique captions for multi-granularity regions. In Stage 1, we construct a mask tree that captures hierarchical relationships between regions. Stage 2 generates short captions based on the parent node. Stage 3 aggregates captions from child nodes, and Stage 4 ensures that each node is assigned a unique caption. Best viewed in zoomed-in.

Previous research has made significant progress in generating dense region captions; however, approaches focusing on multi-granularity regions remain scarce. When considering the granularity of regions, distinguishing their unique attributes becomes crucial (Park & Paik, 2023; Wang et al., 2020b; Liu et al., 2019; Wang et al., 2020a), as visually similar regions frequently appear within an image. Existing approaches have struggled to generate truly unique captions for regions, often producing generic descriptions despite clear visual differences.

This tendency to generate generic captions contradicts human perception, as humans naturally recognize and describe regions based on distinctive attributes like color, position, and shape. However, existing captioning datasets often lack such specificity, and training models on such generic captions that do not emphasize regional uniqueness can contribute to the *mode collapse* problem (Wang et al., 2020b), where models fail to generate diverse and informative captions.

To address this lack of specificity in existing datasets, we propose URECA dataset, a dataset designed to enhance models' ability to generate unique captions for given multi-granularity regions. Our dataset is generated through an automated data pipeline that creates and verifies captions in a stage-wise manner. Specifically, we build our dataset using the publicly available SA-1B dataset, which offers high-resolution images and multi-granularity regions. To further ensure caption quality in the test set, we incorporate a verification step using GPT-4o (OpenAI, 2024) as part of the pipeline.

**Data annotation pipeline.** To generate unique captions that effectively capture multi-granularity, it is crucial to consider both target and non-target regions. Captions that focus solely on the target region often become overly localized and repetitive, making it difficult to distinguish between similar regions. To address this, we structure hierarchical relationships between regions, ensuring that captions incorporate broader contextual information.

At the core of our approach is a mask tree, constructed based on Intersection-over-Union (IoU). This hierarchical structure organizes regions into subset-superset relationships, allowing us to systematically capture dependencies between different regions. This hierarchical structure enables a comprehensive understanding of region dependencies at both global and local levels, ensuring the generation of unique captions. Full implementation details for our data pipeline, including the specific prompts and parameters used to guide the annotation MLLM at each stage, are provided in the Appendix.

This process follows a structured sequence of four stages, as illustrated in Figure 3:

| Dataset | Simple caption | Dense caption | Region caption | Multi-granularity | Unique caption |
|---|---|---|---|---|---|
| RefCOCOg (Yu et al., 2016) | ✓ | ✗ | ✓ | ✗ | ✗ |
| Visual Genome (Krishna et al., 2016) | ✓ | ✗ | ✓ | ✗ | ✗ |
| PACO (Ramanathan et al., 2023) | ✓ | ✗ | ✓ | ✗ | ✗ |
| Partimagenet (Chen et al., 2014) | ✓ | ✗ | ✓ | ✗ | ✗ |
| PRIMA (Wahed et al., 2024) | ✓ | ✓ | ✗ | ✗ | ✗ |
| LLaVA-115K (Liu et al., 2023) | ✓ | ✓ | ✗ | ✗ | ✗ |
| Arcana (Sun et al., 2024) | ✓ | ✓ | ✓ | ✗ | ✗ |
| Osprey (Yuan et al., 2024) | ✓ | ✓ | ✓ | ✗ | ✗ |
| I Dream My Painting (Fanelli et al., 2024) | ✓ | ✓ | ✓ | ✗ | ✗ |
| GRIT (Peng et al., 2023) | ✓ | ✓ | ✓ | ✗ | ✗ |
| LiSA (Lai et al., 2024) | ✓ | ✓ | ✓ | ✗ | ✗ |
| USE (Wang et al., 2024b) | ✓ | ✓ | ✗ | ✓ | ✗ |
| SegCAP (Zhou et al., 2024) | ✓ | ✓ | ✓ | ✓ | ✗ |
| GranD (Rasheed et al., 2024) | ✓ | ✓ | ✓ | ✓ | ✗ |
| DAM (Lian et al., 2025) | ✓ | ✓ | ✓ | ✓ | ✗ |
| **URECA dataset (Ours)** | ✓ | ✓ | ✓ | ✓ | ✓ |

Table 1: **Statistical comparison of previous captioning datasets and URECA dataset in region-level captioning.** The comparison covers different types of captions, including simple captions (*e.g.*, (Ramanathan et al., 2023; He et al., 2022)), dense captions (*e.g.*, (Wahed et al., 2024; Liu et al., 2023)), region captions (*e.g.*, (Yu et al., 2016; Krishna et al., 2016; Sun et al., 2024; Yuan et al., 2024; Wu et al., 2022; Fanelli et al., 2024; Lai et al., 2024)), and multi-granularity captions (*e.g.*, (Wang et al., 2024b; Zhou et al., 2024; Rasheed et al., 2024)). While these datasets provide varying levels of detail, URECA dataset is the only dataset that offers distinctive dense captions and handles multi-granularity regions effectively.

1. **Mask tree generation.** We first construct a mask tree to represent the hierarchical relationships among masks in an image. By comparing the IoU between masks, we can determine their relationships (i.e., superset or subset) within the hierarchy.

2. **Top-down generation.** To ensure that contextual information is effectively incorporated into each node's caption, we generate captions in a top-down manner. In this process, each node refers to its parent node to maintain hierarchical consistency. Specifically, we generate short captions using our annotation MLLM, InternVL2.5-38B (Chen et al., 2024), for each node by referring to captions from the parent node and two types of images that represent the target region: a cropped image of the target region with non-target areas blurred based on the mask (Yang et al., 2023b), and a cropped image of the parent region, where the target region is contoured while non-target areas within the parent region are blurred.

3. **Bottom-up generation.** To ensure that parent nodes have unique captions incorporating relevant details from their child nodes while maintaining contextual coherence, we generate captions in a bottom-up manner. In this process, the parent node refers to its children's captions to generate a more informative and unique caption. Specifically, we aggregate the captions of all child nodes and use our annotation MLLM to generate a refined caption based on the aggregated captions, the parent node's short caption, and an image where the target region is contoured within the full image to preserve its spatial context.

4. **Uniqueness refinement.** To further ensure visually similar regions have distinguishable captions, we introduce a uniqueness refinement process based on image feature similarity using DINOv2 (Oquab et al., 2023). In this stage, similar-looking regions are identified using image features and marked in the image with contours and indexed bounding boxes (Yang et al., 2023a). Our annotation MLLM then generates a unique caption by explicitly differentiating the target region from other visually similar regions.

**Evaluation set.** To ensure the quality of the test dataset when evaluating unique captioning on multi-granularity regions, we additionally implemented a verification stage during the test set generation process. As state-of-the-art MLLMs have demonstrated performance comparable to human annotators' preferences (Lee et al., 2024; Xiong et al., 2024; Ge et al., 2023), we utilized GPT[1], which is widely adopted to simulate human annotators for data generation tasks. Further details about the dataset pipeline can be found in Appendix.

**Data statistics.** We conducted a statistical comparison between previous captioning datasets and URECA dataset. Table 1 highlights their capabilities in region-level captioning. Simple caption

---

[1] gpt-4o-mini-2024-07-18

refers to datasets (Ramanathan et al., 2023; He et al., 2022) that provide basic descriptions, often incorporating object classes in the captions. Dense caption represents datasets (Wahed et al., 2024; Liu et al., 2023) that include multiple attributes, offering more detailed descriptions of the region. Additionally, datasets (Yu et al., 2016; Krishna et al., 2016; Sun et al., 2024; Yuan et al., 2024; Wu et al., 2022; Fanelli et al., 2024; Lai et al., 2024) where captions are explicitly aligned with specific regions fall under the region caption category. As multi-granularity captioning becomes increasingly relevant for real-world applications, recent datasets (Wang et al., 2024b; Zhou et al., 2024; Rasheed et al., 2024) have started to incorporate this aspect. However, none of the existing datasets fully capture all these aspects with captions that describe distinctive attributes of the region while maintaining multi-granularity. Among them, URECA dataset stands out as a unique dataset providing distinct dense captions while effectively handling multi-granularity regions.

## 4 URECA MODEL

The overall architecture of our URECA model is illustrated in Figure 4. Its design is motivated by a central challenge in region-level understanding: how to provide a VLM with a high-fidelity representation of a specific region without compromising the global context of the full image. Existing methods for this task fall short in ways that fundamentally limit their ability to generate unique, multi-granularity captions.

Previous approaches can be broadly categorized. Some encode regions using coordinate-based representations (Liu et al., 2023; Chen et al., 2023), translating bounding boxes into textual tokens. This forces the language model to learn a non-trivial mapping from text to spatial locations and struggles to represent irregularly shaped objects. Other methods rely on image modification, such as overlaying contours (Cai et al., 2024; Shtedritski et al., 2023) or providing a cropped view of the region (Cai et al., 2024). While intuitive, this alters the original visual input, potentially obscuring other objects or contextual cues that are essential for differentiating the target region from similar instances. Recent researches uses feature-level pooling (e.g., RoIAlign) to extract features directly from the visual backbone's feature maps (Wu et al., 2022; Ma et al., 2024; Rasheed et al., 2024). The critical limitation here is that pooling operates on low-resolution feature maps, leading to a significant loss of the fine-grained spatial details, such as precise boundaries and texture that are necessary to distinguish between similar objects and describe regions at varying levels of granularity.

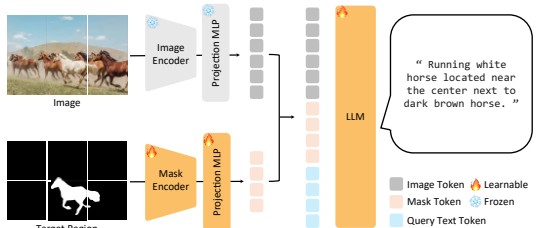

Figure 4: **Architecture of URECA model.** URECA models enables users to generate unique captions that describe distinctive attributes of any region. The mask encoder effectively encodes multi-granularity regions while preserving their identity. The mask token serves as a localizer, guiding the LLM to generate region-specific captions based on the image and query token.

To overcome these limitations, we introduce an architecture founded on the principle of decoupling the region's geometric information from the image's rich visual context. This approach ensures that both data streams are preserved with high fidelity. The core components of our method, which we detail in the following subsections, are a novel processing strategy and a dynamic input mechanism designed for multi-granularity.

### 4.1 DECOUPLED PROCESSING OF REGION AND IMAGE FEATURES

To generate a caption that is not just accurate but also unique, a model must understand both the specific features of a target region and its broader context within the image. Previous approaches alter the image or pooled only for corresponding region features, which irreversibly discards valuable global information and can harm the integrity of the visual features. This loss of context limits the model's ability to reason about object relationships and scene dynamics.

To overcome this limitation, we propose a decoupled processing strategy that preserves the integrity of both the image and the region prompt. Our key insight is to process the region mask and the full image in separate, parallel streams. We introduce a lightweight mask encoder that exclusively

encodes the binary mask into a sequence of feature tokens. These mask tokens act as precise spatial localizers, directing the model's attention without modifying the original image.

The resulting mask tokens are then prepended to the unharmed image tokens from the vision encoder. This simple yet effective approach allows the model to leverage two complementary information sources: the mask tokens provide an unambiguous geometric cue for *where* to look, while the full image tokens provide the rich contextual information for *what* to describe. By doing so, our model effectively utilizes both local and global details to generate captions that are both spatially precise and contextually aware.

Formally, the mask encoding process is:

$$F = \phi(M) \in \mathbb{R}^{N \times D}, \tag{1}$$

where $M \in \{0, 1\}^{H \times W}$ is the input binary mask of height $H$ and width $W$. The mask encoder $\phi(\cdot)$ maps $\mathbf{M}$ to a feature representation $F$, which consists of $N$ spatial tokens in a $D$-dimensional embedding space. Unlike traditional feature pooling, our tokenization approach preserves spatial details, allowing the mask tokens to carry rich information about the region's structure. Full architectural details are provided in the Appendix.

### 4.2 Dynamic Mask Modeling for Multi-Granularity

A key challenge for any fixed-input encoder is handling the diverse scales inherent in multi-granularity region captioning. Resizing a large, high-resolution mask down to a small, fixed input size would inevitably lead to the loss of fine-grained details, defeating the purpose of our high-fidelity approach.

To address this, we propose a dynamic mask modeling, an adaptive tiling strategy that splits the original high-resolution mask into a grid of multiple sub-masks before encoding. Critically, the grid size (e.g., 2x2, 3x3) is adapted based on the original mask's resolution, ensuring that each sub-mask maintains a relatively consistent and high level of detail. This prevents excessive downsampling for large regions and avoids unnecessary padding for small ones. Each sub-mask is then processed by the encoder, and the resulting sequences of tokens are concatenated.

This process splits the original mask $M \in \{0, 1\}^{H \times W}$ into multiple sub-masks $M_{\text{split}}$:

$$M_{\text{split}} = \text{Split}(M) \in \{0, 1\}^{N_s \times H' \times W'}. \tag{2}$$

Here, $N_s$ is the number of sub-masks in the grid. This dynamic approach allows the final mask token sequence length to scale with the input resolution, ensuring a consistently rich and detailed representation. This is particularly crucial for capturing the subtle features that distinguish small or complex objects, making it a cornerstone of URECA model's ability to handle multi-granularity captioning.

## 5 Experiments

### 5.1 Quantitative Results

We report the performance of our URECA model model on URECA dataset as well as previous benchmark datasets (Krishna et al., 2016; Yu et al., 2016). All results are evaluated using an 8B language model trained exclusively on the URECA dataset.

**Unique multi-granularity region captioning.** In Table 2, we present the performance comparison on URECA dataset, a dataset specifically designed to evaluate unique multi-granularity region captions, alongside previous methods. To demonstrate the effectiveness of our approach, we implemented a baseline by running a naïve MLLM (Chen et al., 2024) on URECA dataset. "None" refers to providing the MLLM with only the image, without any explicit region marking. "Contour" refers to marking regions within the image, and "Crop" involves providing the MLLM with a cropped view of the target region. The results indicate that conditioning the MLLM solely on the image or natural language fails to localize regions effectively and generate unique captions.

While previous region-level captioning models (Ma et al., 2024; Huang et al., 2024; Peng et al., 2023; Zhang et al., 2024a;b; Cai et al., 2024) have demonstrated improved performance in generating

Table 2: **Performance comparison of URECA model with baseline methods and previous models on various evaluation metrics**, including BLEU (Papineni et al., 2002), ROUGE (Lin, 2004), METEOR (Banerjee & Lavie, 2005), and BERTScore (Zhang et al., 2019). The results show that URECA model outperforms other methods across all metrics on URECA testset, demonstrating its superior ability to generate unique captions for multi-granularity regions. Note that comparison methods are all trained on URECA dataset.

| Models | BLEU@1 | BLEU@2 | BLEU@3 | BLEU@4 | ROUGE | METEOR | BERTScore | CLAIR |
|---|---|---|---|---|---|---|---|---|
| None | 17.06 | 7.63 | 3.14 | 1.20 | 17.86 | 27.72 | 62.68 | 47.50 |
| Contour | 17.10 | 7.13 | 2.63 | 1.01 | 19.95 | 25.49 | 63.29 | 49.47 |
| Crop | 18.43 | 7.53 | 2.45 | 0.85 | 19.73 | 26.45 | 63.63 | 47.75 |
| GPT-4o | 20.38 | 9.01 | 3.62 | 1.53 | 20.44 | 29.87 | 65.44 | 58.62 |
| SCA | 22.76 | 13.58 | 6.97 | 3.88 | 30.76 | 24.87 | 70.67 | 30.82 |
| KOSMOS-2 | 30.31 | 18.12 | 9.96 | 5.55 | 34.19 | 32.94 | 72.64 | 50.66 |
| Osprey | 31.82 | 20.30 | 12.06 | 7.07 | 36.37 | 34.29 | 73.42 | 53.51 |
| OMG-LLaVA | 34.01 | 21.88 | 13.51 | 8.46 | 38.14 | 37.29 | 74.68 | 29.09 |
| ViP-LLaVA (7B) | 34.17 | 22.07 | 13.96 | 9.00 | 38.17 | 37.68 | 74.62 | 55.94 |
| ViP-LLaVA (13B) | 35.35 | 23.52 | 15.07 | 9.96 | **38.97** | 39.29 | 74.99 | 55.94 |
| **URECA (Ours)** | **39.29** | **23.84** | **15.42** | **9.98** | 38.95 | **41.25** | **75.11** | **66.96** |

unique captions when trained on URECA dataset, they lag behind URECA model either because they struggle to localize multi-granularity regions, alter the original image, or overly constrain the target region without considering the global context.

This underscores that fine-tuning existing captioning models on the URECA dataset enhances their ability to handle multi-granularity captioning. However, URECA model surpasses these approaches by not only generating unique captions across an image but also effectively capturing multi-granularity regions, demonstrating its capability to accurately represent regional information.

**Evaluation of unique captions.** Traditional n-gram-based metrics are not fully equipped to evaluate caption uniqueness. A description's uniqueness can hinge on a single discriminative word, yet conventional metrics treat all words with equal weight, failing to capture this semantic importance. To address this, recent studies have begun to adopt model-based metrics that better assess semantic meaning (Lin et al., 2025; Lian et al., 2025). We therefore provide a comprehensive evaluation using both traditional and semantic-aware metrics (Zhang et al., 2019; Chan et al., 2023), demonstrating that our model achieves state-of-the-art performance in both categories, validating its ability to generate captions that are not only accurate but also uniquely descriptive.

## 5.2 QUALITATIVE RESULTS

Figure 5 provides a qualitative comparison between URECA model and baseline methods, illustrating its superior performance in handling both multi-granularity and uniqueness. In the top example, which tests multi-granularity, baseline models fail to describe the specified region, either describe it as a generic "metal bar" or hallucinating a different scene entirely. In contrast, URECA model accurately describes both the whole object ("pommel horse") and its fine-grained parts ("maroon and metallic legs"), demonstrating its precise localization and descriptive capabilities.

Similarly, in the bottom example focused on multi-granularity, where other baselines failed to locate region. URECA model, however, generates a unique caption by identifying the specific object ("the brown leather boot") and its distinguishing location ("on the man's right foot"). This highlights our model's ability to ground descriptions in the unique visual attributes required by the task. Additional qualitative results are provided in the Appendix.

## 5.3 ABLATION STUDIES

**Effectiveness of mask encoding and dynamic masking.** To evaluate the effectiveness of our proposed methods, we conduct an ablation study by separately implementing each component and assessing their impact on model performance. As presented in Table 3, the baseline MLLM without conditioning

Table 3: **Ablation study of our proposed methods on URECA dataset.**

| Method | ROUGE | METEOR | BERTScore |
|---|---|---|---|
| Baseline | 17.86 | 27.72 | 62.68 |
| + Mask Encoder | 38.46 | 40.72 | 74.73 |
| + Dynamic Mask | 38.95 | 41.25 | 75.11 |

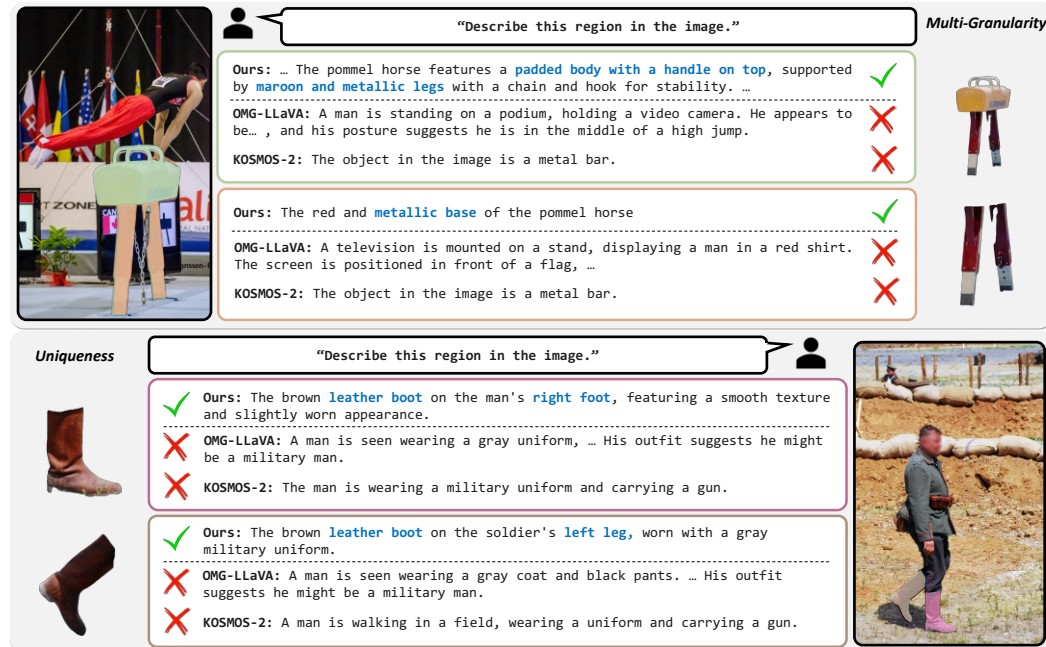

Figure 5: **Qualitative results of the URECA model and comparison models (Peng et al., 2023; Zhang et al., 2024b).** Our model generates unique caption conditioned on multi-granularity regions.

performs poorly. Incorporating our mask encoder, which effectively encodes the target region while preserving its identity, significantly enhances the model's ability to localize regions and generate more descriptive captions. Furthermore, employing our dynamic masking strategy, which divides the original resolution into smaller sub-images, enables the mask encoder to capture finer details of target regions, further improving performance.

**MLLM size.** It is well established that performance improves with larger foundation models (Li et al., 2024; Chen et al., 2024; Zhang et al., 2022; Bai et al., 2025), as their knowledge capacity scales with model size. Our URECA model follows this trend, achieving better performance as its size increases, as shown in Table 4. While the 1B model records the lowest performance, the largest model (8B) achieves the highest.

Table 4: **Ablation study on model size.**

| Model Size | ROUGE | METEOR | BERTScore |
|---|---|---|---|
| 1B | 32.00 | 33.99 | 71.77 |
| 2B | 36.64 | 39.00 | 73.92 |
| 4B | 36.58 | 38.75 | 73.97 |
| 8B | 38.95 | 41.25 | 75.11 |

**Mask token length.** We demonstrated that our mask encoder effectively captures regions while preserving their identity. To analyze the impact of the number of tokens generated by the mask encoder, we conduct an ablation study, as shown in Table 5. We investigate the effect of increasing the number of mask tokens. As the number of tokens increases,

Table 5: **Ablation study on mask token length.**

| Token Length | ROUGE | METEOR | BERTScore |
|---|---|---|---|
| 4 | 35.44 | 38.01 | 73.51 |
| 8 | 37.06 | 38.50 | 74.21 |
| 16 | 38.95 | 41.25 | 75.11 |

the representation becomes more detailed, allowing for finer details to be captured, particularly in smaller regions.

## 6 CONCLUSION

We present URECA dataset, a regional captioning dataset that includes multi-granularity regions. Our primary objective is to annotate regions with unique captions that exclusively describe the target region. To achieve this, we propose an automated data pipeline that generates distinctive captions using a mask tree, which captures the hierarchical relationships between regions. To ensure high-quality evaluation, we introduce a verification stage to validate the test set. Furthermore, we introduce URECA model, which encodes masked regions while effectively preserving their identity. To retain finer details, we propose dynamic masking, leveraging the LLM's flexible input length to encode masks even in high-resolution views.

## REPRODUCIBILITY STATEMENT

We detail the training configurations in Appendix A. We will also release our code and model check-points to ensure reproducibility.

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

# APPENDIX

## A    IMPLEMENTATION DETAILS

We leverage InternVL-2.5 (Chen et al., 2024) along with our mask encoder, which consists of convolutional layers followed by a two-layer MLP as the projection layer for mask tokens. For our experiments, we set the mask token length to 8. While our ablation study (Table 6) indicates that performance continues to improve with 16 tokens, we selected a length of 8 to maintain a favorable balance between descriptive performance and the computational cost associated with longer token sequences during training. The input to the mask encoder is resized to 448x448, and the dimension of the mask tokens matches the feature dimension of the MLLM.

We train our model on four Tesla A100 GPUs (40GB) using LoRA (Hu et al., 2021). Specifically, training is conducted in two stages: first, we train the mask encoder and projection layer, followed by LoRA fine-tuning of the MLLM. We use a batch size of 16 for LoRA tuning.

For evaluation, we adopt standard metrics from prior work, including BLEU (Papineni et al., 2002), ROUGE (Lin, 2004), and METEOR (Banerjee & Lavie, 2005). While these metrics allow for direct comparison, they are not designed to measure descriptive uniqueness, which is the primary goal of our research. A more detailed discussion on these limitations is provided in the Appendix H. To better assess semantic quality, we supplement these scores with BERTScore (Zhang et al., 2019) and the CLAIR score (Chan et al., 2023).

### A.1    MASK ENCODER ARCHITECTURE

Our mask encoder is a lightweight convolutional network designed to transform a binary region mask into a sequence of feature tokens. The architecture is intentionally kept simple to ensure efficiency and reproducibility. As detailed in Algorithm 1, the encoder consists of two sequential 2D convolutional layers. Each layer uses a 3x3 kernel, a stride of 2, and padding of 1, effectively downsampling the input by a factor of 2 at each step. A ReLU activation function follows each convolution. The resulting feature map is flattened and then projected to the MLLM's hidden dimension using a two-layer MLP, which serves as the projection head. All convolutional and linear layers in the mask encoder are initialized using the Xavier normal initialization method.

---

**Algorithm 1** Mask Encoder Pseudo-Code

---

**Require:** Binary mask $M$ of size $H \times W$
**Ensure:** Mask tokens $F$ of size $N \times D$ (where $N = 8$ is the number of tokens and $D$ is the LLM hidden size)
1: **function** ENCODEMASK($M$)
2:     $x \leftarrow \text{reshape}(M, [1, 1, H, W])$         ▷ Reshape mask to have a channel dimension

3:     $x \leftarrow \text{Conv2d}(x, \text{in\_channels}=1, \text{out\_channels}=C, \text{kernel}=3, \text{stride}=2, \text{padding}=1)$
4:     $x \leftarrow \text{ReLU}(x)$                                              ▷ First convolutional block

5:     $x \leftarrow \text{Conv2d}(x, \text{in\_channels}=C, \text{out\_channels}=C, \text{kernel}=3, \text{stride}=2, \text{padding}=1)$
6:     $x \leftarrow \text{ReLU}(x)$                                          ▷ Second convolutional block

7:     $x \leftarrow \text{flatten}(x)$                                        ▷ Flatten the spatial dimensions

8:     $x \leftarrow \text{MLP}(x, \text{out\_features}=D)$         ▷ Project to LLM hidden dim via 2-layer MLP

9:     **return** $x$
10: **end function**

---

## B    ADDITIONAL RELATED WORK

Large Language Models (LLMs) have demonstrated pioneering performance in instruction following capabilities, integrating diverse knowledge from extensive datasets, and performing complex

reasoning tasks. However, a significant limitation of LLMs is their reliance solely on natural language inputs. To address this, LLaVA (Liu et al., 2023) was the first to explore the integration of image and text modalities by representing visual features as visual tokens. Building upon this, models such as Flamingo (Alayrac et al., 2022) and BLIP-2 (Li et al., 2023) have further advanced Multimodal Large Language Models (MLLMs) by incorporating powerful visual backbones. These models effectively bridge the two modalities and have shown strong performance in tasks like image captioning and visual question answering. Building on these advancements, recent efforts have aimed to extend these models to handle more complex tasks, including reasoning over segmentation (Lai et al., 2024; Ren et al., 2024), optical character recognition (Wang et al., 2024a; Dong et al., 2024), and grounding (Plummer et al., 2016; Rasheed et al., 2024; Wang et al., 2024b; Zhou et al., 2024; Halbe et al., 2024).

## C  LIMITATIONS

While our mask encoder effectively encodes multi-granularity regions without losing details, localizing the region in a sequential manner may occasionally cause the MLLM to misidentify the target region. Since we do not explicitly constrain target regions using image features or direct markers, the localization signal provided to the MLLM may be weaker compared to previous methods. Enhancing region encoding by incorporating both the mask and additional image features, rather than relying solely on sequential conditioning, could improve the MLLM's ability to accurately localize the target region.

## D  REGION-LEVEL CAPTIONING

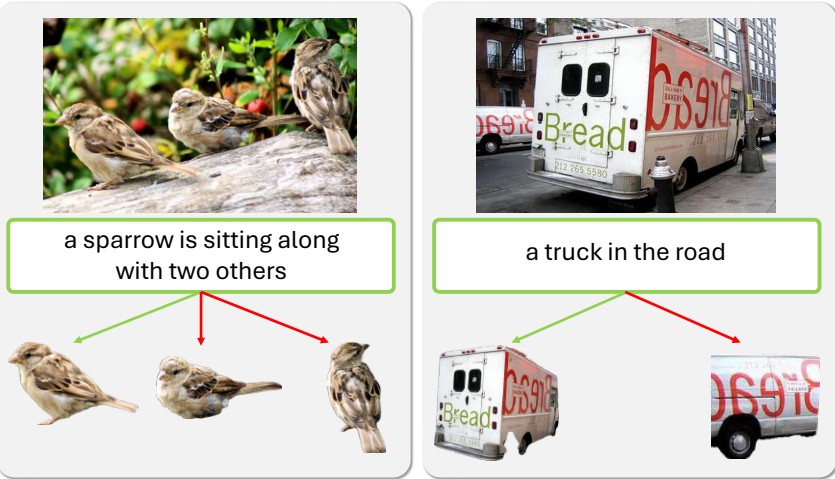

Figure A: **Qualitative examples from the RefCOCOg dataset.** The green arrows indicate the **ground-truth annotation in the validation set**, while the red arrow highlights another **possible candidate** that can be mapped to the caption.

In Table A, we present the zero-shot performance of URECA model on RefCOCOg (Yu et al., 2016) and Visual Genome (Krishna et al., 2016). On RefCOCOg, URECA model demonstrated competitive performance, while on Visual Genome, it achieved state-of-the-art results compared to previous approaches.

Notably, unlike prior methods, URECA model achieves these results without using the benchmarks' training sets, highlighting the strong generalization ability of URECA dataset. This

Table A: **Quantitative results on region-level captioning task.** Performance comparison on the METEOR for the RefCOCOg (Yu et al., 2016) and Visual Genome (Krishna et al., 2016) datasets.

| Models | RefCOCOg | Visual Genome |
|---|---|---|
| ControlMLLM (Wu et al., 2024) | 14.0 | - |
| Kosmos-2 (Peng et al., 2023) | 14.1 | - |
| GRiT (Wu et al., 2022) | 15.2 | 17.1 |
| SLR (Yu et al., 2017) | 15.9 | - |
| GLaMM (Rasheed et al., 2024) | 15.7 | 17.0 |
| OMG-LLaVA (Zhang et al., 2024b) | 15.3 | - |
| ViP-LLaVA (Cai et al., 2024) | 16.6 | - |
| Groma (Ma et al., 2024) | 16.8 | 16.8 |
| RegionGPT (Guo et al., 2024) | 16.9 | 17.0 |
| Omni-RGPT (Heo et al., 2025) | 17.0 | 17.0 |
| Draw-and-Understand (Lin et al., 2024) | **23.9** | - |
| **URECA (Zero-Shot)** | 16.1 | **18.4** |

suggests that URECA dataset covers diverse region granularities with well-aligned captions, enabling better regional understanding. By effectively learning from a dataset with varying granularities, URECA model effectively localizes and generates captions across different scales, making it highly adaptable to region-level captioning even on the zero-shot tasks.

It is important, however, to acknowledge a fundamental distinction in the evaluation. As illustrated in Figure A, datasets such as RefCOCOg and Visual Genome do not enforce unique annotations for each region. A single area—like the truck shown—can be described with a general caption ('a truck in the road') or a more specific one. This inherent ambiguity means that evaluating on these benchmarks cannot be seen as the same task as generating a single, uniquely identifying caption. Despite this misalignment, the fact that URECA model achieves such a **comparable performance** is particularly noteworthy. It underscores the model's robustness, proving its ability to generate high-quality, relevant descriptions even when the evaluation criteria are broader and less constrained than our primary objective.

## E   MORE QUALITATIVE RESULTS

We visualize more qualitative results of URECA model with previous apporaches (Cai et al., 2024; Zhang et al., 2024b) in Figure B.

## F   DATASET VISUALIZATION

We provide visual examples of our dataset to illustrate its diversity and complexity. Figure C showcases representative samples, highlighting key variations in object appearance, background context, and challenging scenarios. For optimal viewing, we recommend zooming in and viewing the figures in color to better observe fine details.

## G   DATA PIPELINE

To generate unique regional captions with multi-granularity, we propose a structured four-stage process:

**Stage 1: Mask Tree Construction.**   We first build a mask tree for each image using masks from the SA-1B dataset (Kirillov et al., 2023). Intersection over Union (IoU) between masks is computed to determine containment relationships. Each tree has a root node representing the entire image, with subsequent nodes structured hierarchically based on these containment relationships.

**Stage 2: Top-Down Caption Generation.**   In this stage, we identify primary nodes directly under the root node, termed *main objects*, whose depth exceeds a predefined threshold. Short captions are then hierarchically generated from these main objects downward through descendant nodes. Each node creates concise captions using contextual information from parent and sibling nodes to maintain coherence and uniqueness. Specific prompts used in this step are detailed in Table B.

**Stage 3: Bottom-Up Caption Refinement.**   Short captions generated in Stage 2 are expanded into detailed descriptions. Each node enriches its caption by incorporating information from child nodes, ensuring hierarchical consistency and comprehensive context. Prompts for this refinement stage are provided in Table C.

**Stage 4: Uniqueness Refinement.**   Finally, captions are refined by evaluating visual similarity between regions using DINO v2 (Oquab et al., 2023). Regions with high visual similarity have their captions adjusted by emphasizing distinguishing features, maintaining semantic relevance and uniqueness. Prompts for uniqueness refinement are described in Table D.

Through these stages, we systematically generate multi-granularity captions that accurately describe each region with clarity, context, and uniqueness in an automated manner.

## H  DISCUSSION

Evaluating unique caption generation for regional captioning tasks using traditional metrics such as BLEU (Papineni et al., 2002), METEOR (Banerjee & Lavie, 2005), ROUGE (Lin, 2004), and CIDEr (Vedantam et al., 2015) presents inherent limitations. These metrics primarily assess similarity to reference captions based on n-gram overlap, without distinguishing between essential and non-essential words. However, in unique captioning, it is crucial to generate descriptions that highlight distinctive attributes, ensuring that the caption effectively differentiates the target region from others. Existing evaluation methods treat all words equally, failing to account for the importance of discriminative terms. As a result, captions that successfully emphasize key distinguishing features may not receive high scores if their phrasing deviates from reference texts, even if they better serve the task's objective. This limitation suggests the need for alternative evaluation approaches that better capture the quality and distinctiveness of unique captions.

## I  LIMITATION

Our work relies on a fully automated pipeline for dataset creation and evaluation, and as such, does not include a large-scale human study to validate the perceived quality and uniqueness of the captions. While we use GPT-4o for test set verification, which has shown strong correlation with human preferences in prior work, we acknowledge that direct human evaluation remains the gold standard. We believe this is a necessary trade-off for the scale of our dataset, and we identify rigorous human studies as a critical direction for future work.

## J  METHODOLOGICAL JUSTIFICATION FOR DATASET CURATION

A potential concern regarding our dataset creation methodology is the use of two models from the InternVL family (8B for the training set, 38B for the test set), which could be perceived as an unfair evaluation setting. However, we argue that this approach is methodologically sound, does not confer an unfair advantage to our model, and aligns with state-of-the-art practices. Our justification is threefold: (1) the models are architecturally and functionally distinct, positioning the larger model as a valid "annotation oracle"; (2) the capability gap between the models is substantial and supported by established theoretical principles; and (3) the methodology aligns with broader trends in scalable, model-driven data generation.

### J.1  ARCHITECTURAL AND TRAINING HETEROGENEITY

The InternVL 8B and 38B models are not merely scaled versions of one another but are heterogeneous compositions featuring significant architectural and training divergences. This compositional difference provides a strong argument against the notion of a homogenous model family.

- **Distinct LLM Backbones:** Models at different scales within the InternVL series often incorporate Large Language Model (LLM) backbones from entirely different developers. For instance, the `InternVL2.5-8B` model utilizes the `internlm2_5-7b-chat` LLM, whereas the `InternVL2.5-38B` model is built upon the `Qwen2.5-32B-Instruct` LLM (ModelScope, 2024). These LLMs are developed by separate organizations with unique architectures, training datasets, and alignment philosophies, resulting in fundamentally different internal knowledge representations and inductive biases.

- **Asymmetric Application of Advanced Training:** The larger models in the InternVL family are subjected to more advanced and qualitatively different training paradigms designed to enhance reasoning and coherence. Techniques such as Mixed Preference Optimization (MPO) and Cascade Reinforcement Learning (RL) are asymmetrically applied, creating a significant capability gap (Wang et al., 2025; Zhu et al., 2025). For example, fine-tuning with MPO yields a 4.5-point improvement on multimodal reasoning benchmarks for the

`InternVL3-38B` model, a gain attributed primarily to the training algorithm itself rather than the data (Zhu et al., 2025). This "specialized education" endows the 38B model with a more robust and human-aligned reasoning process that is qualitatively distinct from the 8B model.

## J.2 CAPABILITY GAP AND ORACLE-BASED ANNOTATION

The architectural and training differences result in a substantial capability gap, which is consistent with established principles of AI scaling.

- **Neural Scaling Laws:** A large body of empirical research has demonstrated that model performance improves predictably as a power-law function of model parameters, dataset size, and compute (Kaplan et al., 2020; Hoffmann et al., 2022; Bahri et al., 2021). The nearly five-fold increase in parameter count from 8B to 38B is expected to yield a significant, non-linear improvement in performance, justifying the use of the larger model as a higher-quality source of ground-truth labels.

- **Emergent Abilities:** It is well-documented that capabilities can be absent in smaller-scale models but appear abruptly in larger-scale models (Wei et al., 2022; Sciacca et al., 2025). Complex, multi-step reasoning, a prerequisite for high-quality region captioning, is precisely the type of task where such emergent abilities manifest. It is therefore highly plausible that the 38B model possesses sophisticated compositional understanding and reasoning skills that are fundamentally non-existent in the 8B model.

Due to this significant capability gap, our methodology should be understood as **oracle-based annotation** rather than a form of data contamination (Balloccu et al., 2025; Holistic AI, 2024). The test set generated by the 38B "oracle" represents a target distribution of quality and complexity that a model trained on data from the much weaker 8B model cannot trivially replicate. The evaluation, therefore, remains a challenging and fair test of the model's ability to generalize towards the capabilities of a far more powerful system.

## J.3 ALIGNMENT WITH STATE-OF-THE-ART METHODOLOGIES

Our approach follows established and peer-reviewed procedures for scalable data creation and evaluation in vision-language research.

- **The "LLM-as-a-Judge" Paradigm:** Our methodology is a logical extension of the widely accepted "LLM-as-a-Judge" framework, where powerful models like GPT-4 are used as scalable proxies for human evaluators (Zheng et al., 2023; Liu et al., 2024; Fu et al., 2024). The principle that a more capable model can reliably assess the quality of a less capable one has been validated in numerous studies, with LLM-human agreement rates often exceeding 80% (Zheng et al., 2023). If a model is trusted to *judge* quality, it can certainly be trusted to *generate* high-quality annotations.

- **Synthetic Data Generation:** The use of generative models to create training and evaluation data is a rapidly growing trend across vision-language research to overcome the bottleneck of manual annotation. Frameworks like SynGround have shown that training on purely synthetic data can significantly improve visual grounding performance (He et al., 2024). Similarly, projects like Cap3D use a pipeline of pretrained models to generate high-quality descriptive text for 3D objects at a scale that would be infeasible with human annotation (Wu et al., 2023). Our approach fits squarely within this modern paradigm of leveraging powerful foundation models for scalable dataset creation.

In summary, the significant architectural, training, and capability differences between the InternVL 8B and 38B models, combined with the alignment of our methodology with broader state-of-the-art practices, provide a robust justification for our dataset curation strategy. This approach does not create an unfair evaluation setting but rather employs a more powerful, distinct, and validated annotation oracle to establish a high-quality and challenging benchmark.

# K    USE OF LARGE LANGUAGE MODELS

In accordance with the ICLR 2026 submission policy, we disclose that Large Language Models were used to assist in grammar correction and polishing of the writing in this paper.

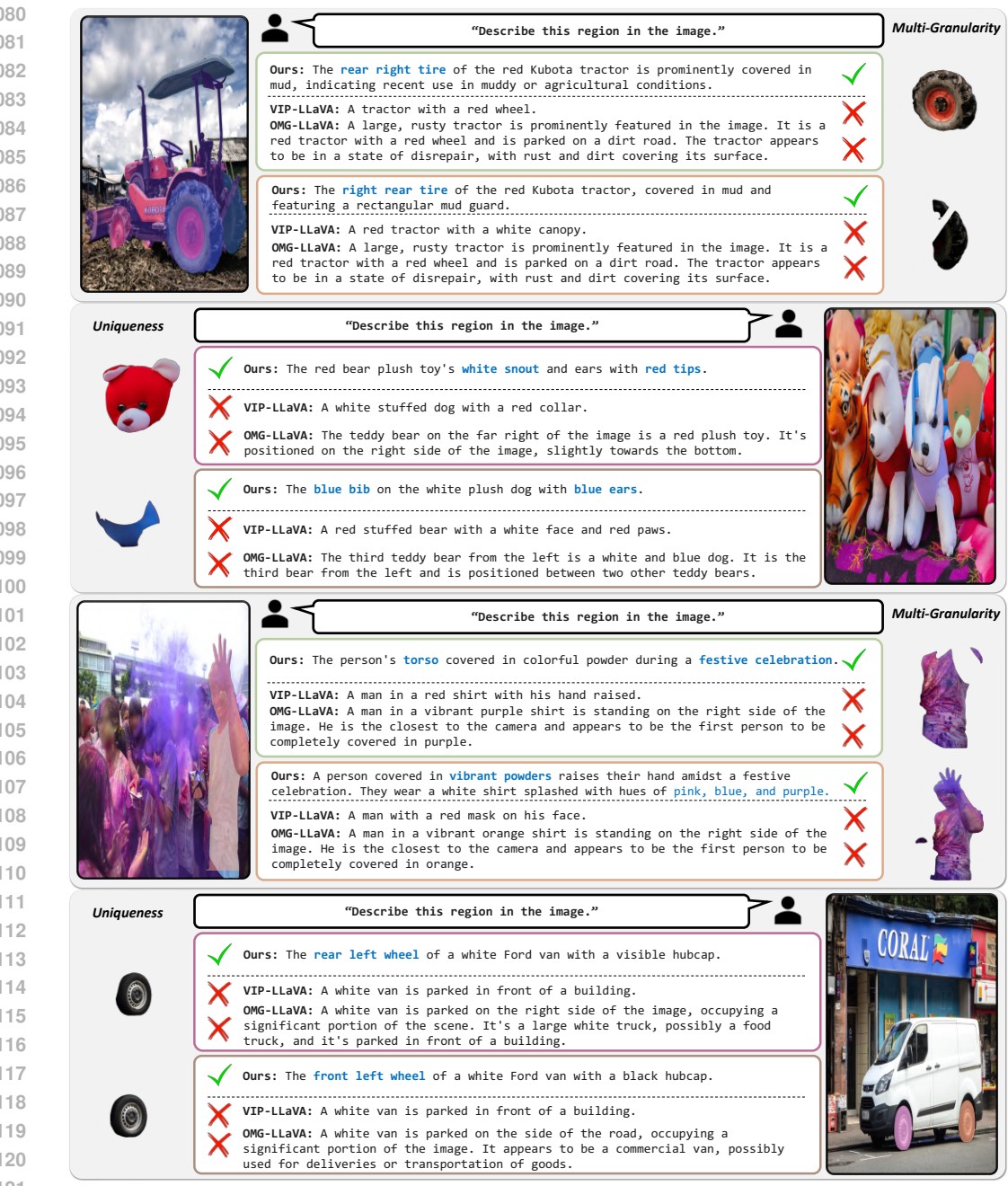

Figure B: **Qualitative results of the URECA model and comparison models Cai et al. (2024); Zhang et al. (2024b).** Our model generates unique caption conditioned on multi-granularity regions.

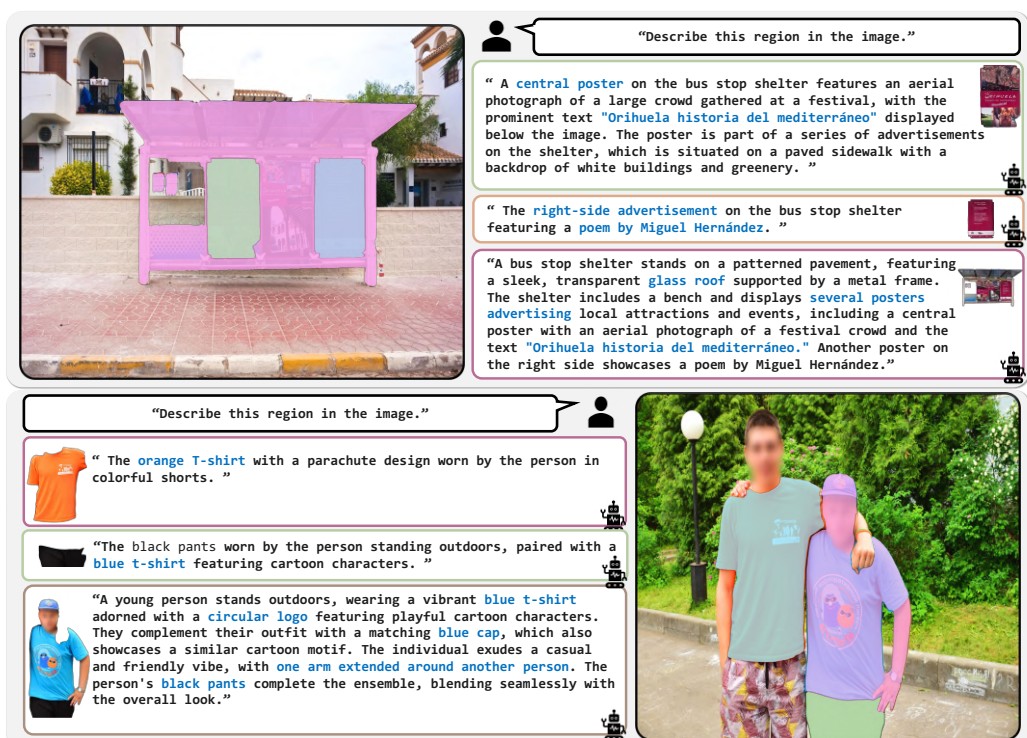

Figure C: **Example data generated by our data curation pipeline.**

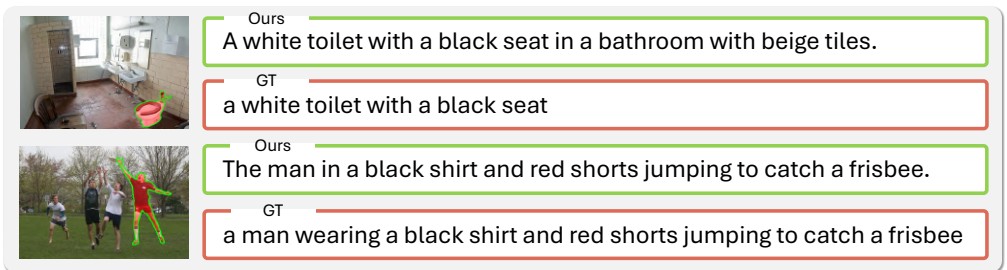

Figure D: Qualitative results of our URECA model on the RefCOCOg Yu et al. (2016) dataset.

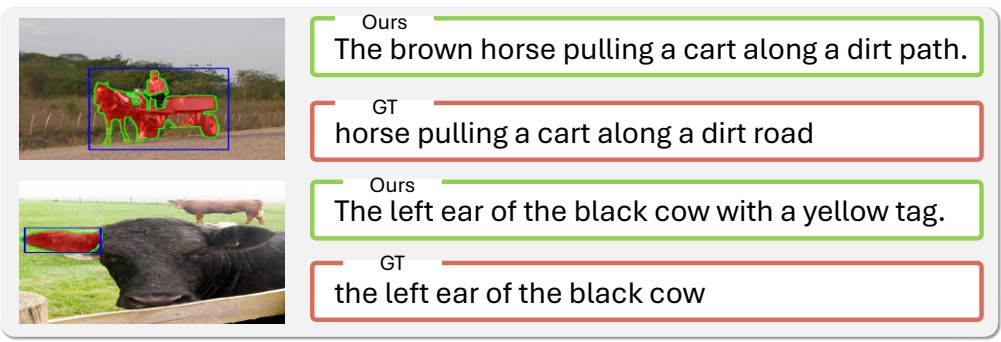

Figure E: Qualitative results of our URECA model on the Visual Genome Krishna et al. (2016) dataset.

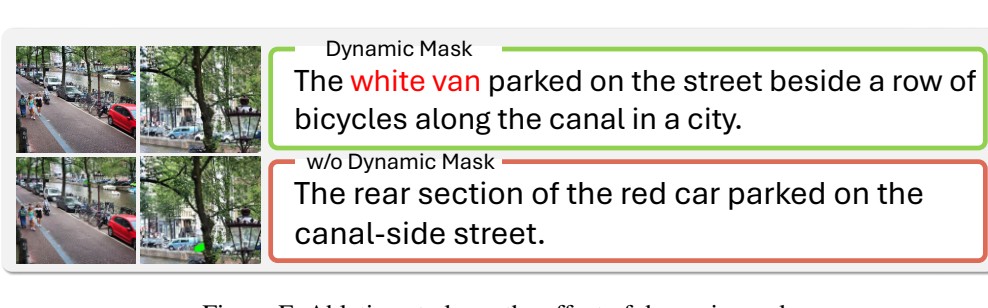

Figure F: Ablation study on the effect of dynamic mask.

```
<task>
    You are a detailed caption generator tasked with describing the main object
    in images. Your goal is to create a simple phrase that accurately represents
     the main object while avoiding hallucination.
</task>
<objectives>
    1. The main object is a subpart of a larger object; therefore, the main
    object alone may provide insufficient information.
    2. The primary focus of the caption must be on the main object while also
    considering its positional relationship or functional connection with the
    larger object.
    3. The primary focus of the caption must be on the main object, emphasizing
    attributes like color, texture, shape, and action if visible.
    4. The background is blurred to emphasize the main object. Focus solely on
    describing the main object in detail without mentioning the blurred
    background.
    5. The caption should be distinguishable from other subparts of the same
    larger object so that the region can be identified solely by looking at the
    caption. Therefore, the caption should incorporate positions or attributes
    that are unique to the main object.
    6. Creating a unique caption is important, but the most critical aspect is
    accuracy. Do not add unnecessary information solely for the sake of
    uniqueness.
</objectives>
<inputDetails>
    1. Image-1 highlights the main object with a yellow contour to illustrate
    its relationship with the larger object.
    2. Image-2 shows the main object cropped from the larger object.
    3. A description of the larger object will be provided in the prompt to help
     identify the main object.
    4. Descriptions of other subparts of the same larger object will also be
    provided. The caption for the main object must be clearly distinguishable
    from the descriptions of these subparts.
</inputDetails>
<descriptionOfLargerObject>
    "Description from the parent object"
</descriptionOfLargerObject>
<descriptionOfSubparts>
    "Descriptions from objects on the same level, if present."
</descriptionOfSubparts>
<outputFormat>
    1. Provide a simple phrase focusing on the main object while considering its
     positional relationship or functional connection with the larger object.
    2. The larger object may contain another object with similar attributes to
    the main object. The caption should be written in a way that clearly
    distinguishes the main object from these similar objects.
    3. Keep the caption concise, limiting it to one sentence while ensuring
    clarity and coherence.
    4. Do not explicitly mention the yellow contour or its presence in the image
    .
    5. Use contextual information from Image-1 to describe the main object's
    relationship with the larger object, while referencing its attributes from
    Image-2.
    6. Contextual details from Image-1 and the description of the larger object
    should be used only to support the description of the main object.
</outputFormat>
<outputExamples>
    "8 in-context examples"
</outputExamples>
```

Table B: Prompts for top-down generation. Captions are generated hierarchically from main objects to descendants while ensuring contextual coherence and uniqueness.

```
<task>
    You are a detailed caption generator tasked with describing the
    main object in images.
    Your goal is to create precise and detailed captions while avoiding
     hallucination.
</task>
<objectives>
    1. The caption must primarily focus on the main object while
    considering its
        contextual information to clearly identify what it is.
    2. The caption must emphasize the main object's attributes, such as
     color, texture, shape, and action if visible.
    3. Describe only what is visible in the image. Avoid adding any
    information that is not present.
    4. The main object is highlighted with a yellow contour.
    5. A short description of the main object will be provided in the
    prompt, which can be used to describe the main object.
    6. The main object consists of multiple subparts, and descriptions
    of these subparts will be provided in the prompt.
    7. The description of subparts may contain inaccurate, unimportant,
     or redundant information. Use only the essential details that do
    not contradict the given image to ensure that the caption for the
    main object compositionally reflects relevant information from these
     subparts.
</objectives>
<inputDetails>
    1. An image with the main object marked by a yellow contour will be
     provided.
    2. A short description of the main object will be included in the
    prompt.
    3. Descriptions of the subparts of the main object will also be
    provided in the prompt.
</inputDetails>
<descriptionOfMainObject>
    "Description from the main object."
</descriptionOfMainObject>
<descriptionOfSubparts>
    "Descriptions from the child objects, if present."
</descriptionOfSubparts>
<outputFormat>
    1. Provide a single descriptive paragraph that focuses on the main
    object.
    2. Do not use bullet points or lists.
    3. Incorporate details from the provided descriptions to accurately
     depict the main object.
    4. Never mention the presence of the yellow contour in any form.
    5. Structure the caption clearly and concisely, avoiding excessive
    detail or verbosity. Do not start with phrases like "The image shows
    ...".
    6. Ensure the focus is evident without explicitly stating that it
    is the main object.
</outputFormat>
```

Table C: Prompts for bottom-up generation. Captions are refined by incorporating child node information to maintain hierarchical consistency.

```
<task>
    You are a caption refinement model that enhances given descriptions
     to generate unique and precise captions for objects in an image.
    Your goal is to refine the provided caption based on contour-based
    indexing while maintaining clarity and specificity.
</task>
<objectives>
    1. Describe only what is visible in the image. Avoid adding any
    information that is not present.
    2. The image contains multiple contours in different colors, each
    with a corresponding index, marking distinct objects.
    3. The main object corresponds to index 0 and is specifically
    outlined with a blue contour.
    4. Your task is to refine the caption for index 0, highlighting its
     unique attributes while clearly differentiating it from other
    indexed contours in the image.
    5. The refined caption must primarily focus on index 0 while
    considering its contextual information to clearly identify it from
    other indices.
    6. The caption must emphasize index 0's attributes, such as color,
    texture, shape, and action, to make caption unique.
</objectives>
<inputDetails>
    1. The contours in the image are color-coded, and each contour has
    a corresponding index.
    2. The index corresponding to each contour is placed at the center
    of the contour, matching its color.
    3. The initial caption for index 0 (blue contour) is provided as
    input.
    4. The refined caption should ensure the distinction between index
    0 (blue contour) and other objects in the image.
</inputDetails>
<refinementGuidelines>
    1. Preserve the core meaning of the given caption while improving
    its specificity and uniqueness.
    2. Emphasize key attributes that differentiate index 0 (blue
    contour) from other indices.
    3. Avoid mentioning the presence of contours or annotations
    explicitly in the caption.
    4. Keep the refined caption clearly yet descriptive.
    5. Ensure that the final caption remains a natural, human-like
    description of the object.
    6. Do not use bullet points or lists.
    7. Do not start the answer with words like "Certainly!".
</refinementGuidelines>
<captionForIndex0>
    "Description from the target (index 0) object"
</captionForIndex0>
<outputFormat>
    1. Provide a single descriptive paragraph that maintains clarity
    and coherence focusing on index 0 (blue contour)
    2. The refined caption should distinguish index 0 (blue contour)
    from other indices.
    3. Avoid generic or ambiguous descriptions.
    4. The refined caption should make index 0 clearly stand out from
    the other indexed objects without using phrases like "distinguished
    by" or similar expressions.
    4. Do not reference the contour colors or indices directly.
</outputFormat>
```

Table D: Prompts for uniqueness refinement. Captions are refined by distinguishing visually similar regions while preserving semantic relevance.

