# OpenReview forum: "URECA: Unique Region Caption Anything"
_ICLR.cc/2026/Conference — ICLR 2026 Conference Withdrawn Submission_

### Official Review · Reviewer_nrr6 · 2025-10-26

**Soundness:** 2
**Presentation:** 3
**Contribution:** 2
**Rating:** 4
**Confidence:** 4

**Summary:**

This paper addresses a significant and often-overlooked problem in region captioning: the lack of "uniqueness" in generated descriptions. Current models frequently produce generic labels that could apply to multiple regions within the same image, limiting their utility. To address this, the authors make two primary contributions:

The URECA Dataset: A new, large-scale benchmark dataset designed to enforce a one-to-one mapping between a visual region and a descriptive caption. This is constructed via a novel four-stage automated pipeline that includes a "Uniqueness Refinement" step to explicitly differentiate between visually similar instances.

The URECA Model: A model architecture featuring two innovations: (a) a "decoupled processing" strategy that uses a separate, lightweight mask encoder to process the region prompt while preserving the full image context, and (b) "dynamic mask modeling" to handle multi-granularity inputs by tiling high-resolution masks.

The authors demonstrate their model's effectiveness on their new dataset and show strong zero-shot generalization on standard benchmarks like Visual Genome.

**Strengths:**

* Important and Well-Defined Problem: The paper clearly articulates the critical need for "caption uniqueness." The failure of current VLMs to distinguish between similar instances (e.g., "the instructor with the red beanie in the back" vs. "the instructor with the black beanie in front") is a real and important limitation for many downstream applications, such as referring segmentation or fine-grained human-computer interaction. The motivation is strong and well-supported.
* Novel and Valuable Dataset Contribution: The four-stage data curation pipeline, particularly the "Uniqueness Refinement" stage (Stage 4), is a clever piece of data engineering. Using DINOv2 to identify visually confusable regions and then prompting an MLLM to generate distinguishing descriptions is an effective and scalable method. This dataset, by itself, appears to be a valuable contribution to the community, directly targeting a weakness not explicitly addressed by prior benchmarks.

**Weaknesses:**

Despite the clear motivation and dataset contribution, the paper's claims regarding its model contribution are unsubstantiated due to a severe lack of comparison with critical baselines and related work.

1. The URECA Model Lacks Comparison to SOTA Mask Injection Techniques: The authors propose their decoupled mask encoder as a key architectural innovation. However, the field has recently seen significant work on how to best inject region-based prompts (like masks) into VLMs. This paper fails to compare against any of them.

* FINECAPTION [CVPR 2025] [1]: This work also focuses on compositional, multi-granularity captioning. How does its method for handling fine-grained region inputs compare, quantitatively, to URECA's simple mask encoder?
* AlphaCLIP [CVPR 2024] [2]: This paper explores several sophisticated strategies for injecting mask information into CLIP. URECA's approach of encoding the mask to tokens and prepending them to the image features is one possible strategy, but it is not compared to the fusion or adapter-based methods explored in AlphaCLIP. Without a rigorous quantitative comparison against these state-of-the-art methods (trained on the URECA dataset for a fair comparison), it is impossible to validate whether the URECA model's architecture is truly novel, efficient, or effective. As it stands, the model contribution is unverified.

2. Omission of Key Model and Dataset Baselines: The paper's evaluation in Table 2 and dataset comparison in Table 1 feel incomplete and potentially "cherry-picked" by omitting highly relevant and impactful recent work.
* GLAMM: This is a state-of-the-art model for pixel-level grounding and multi-granularity description. It is a critical baseline. While it is mentioned in a zero-shot comparison in the Appendix (Table A), it must be trained on the URECA dataset and included as a primary baseline in Table 2. Its omission from the main results table is a major flaw that undermines the paper's SOTA claims.
* DenseWorld-1M [3] Dataset: The authors claim in Table 1 that existing datasets lack multi-granularity or dense region captions, yet they fail to mention or discuss DenseWorld-1M. This recent dataset provides 10 million dense grounded captions. The authors must explicitly argue why DenseWorld-1M is insufficient and how URECA's "uniqueness" objective differs from DenseWorld-1M's "density" objective. Without this differentiation, the necessity of the URECA dataset itself is called into question.

[1] FINECAPTION: Compositional Image Captioning Focusing on Wherever You Want at Any Granularity

[2] Alpha-CLIP: A CLIP Model Focusing on Wherever You Want

[3] DenseWorld-1M: Towards Detailed Dense Grounded Caption in the Real World

**Questions:**

This paper introduces an important problem (caption uniqueness) and presents a promising dataset (URECA) with a novel curation pipeline. However, the paper's second major contribution, the URECA model, is not validated against relevant SOTA mask-injection techniques (e.g., FINECAPTION, AlphaCLIP). Furthermore, the experimental evaluation is weakened by the omission of a critical model baseline (GLAMM) and the failure to situate the dataset contribution against other recent large-scale captioning datasets (DenseWorld-1M).

Due to these significant gaps in the experimental validation, I am voting for Borderline Reject. The authors must provide these critical comparisons to substantiate their claims of model novelty and SOTA performance.

**Details Of Ethics Concerns:**

no concerns

---

### Official Review · Reviewer_b6vn · 2025-10-30

**Soundness:** 2
**Presentation:** 1
**Contribution:** 2
**Rating:** 2
**Confidence:** 4

**Summary:**

This paper introduces a dataset which is for region captioning tasks. Its key contribution lies in the uniqueness when facing some objects with similar appearance descriptions. This paper also introduces an architecture to better mix images with their region masks and further build a URECA model.

**Strengths:**

- This paper provides a novel four-stage automated pipeline for dataset construction. The mask-tree method is interesting and well-modeled.
- Using the decoupled idea in image region tasks and using dynamic mask modeling can directly solve the lossy encoding and some limitations of prior VLMs.

**Weaknesses:**

- The core goal of this work is to eliminate inaccurate recognition caused by duplicate descriptions, but the paper fails to demonstrate whether this method fundamentally solves the problem. It remains unclear whether a valid one-to-one sequence can still be established for two individuals/objects with similar appearances and maybe the same gender—this is a critical scenario to verify the method’s effectiveness.
- This paper adopts a pairwise matching approach to fine-tune descriptions, but it does not provide sufficient evidence (e.g., ablation studies on noisy inputs or variant cases) to prove the robustness of this strategy when there are a large number of similar objects.
- This paper does not clarify the criteria for determining whether content at multiple granularities should be regarded as the same object. Small grass can be used in mask tree, but could overly small materials lead to the possibility of the dataset being contaminated?
- For dataset-focused work, a key way to demonstrate the dataset’s value is to reproduce works that use relevant datasets and thereby prove the proposed dataset’s advantages in performance or functionality. However, this critical validation step is absent from the paper. The key contribution in this paper is "Unique Caption" while it does not provide empirical evidence to confirm whether this component actually gains such a significant effect on improving performance.
- Model implementation details that are essential for result reproducibility and credibility, are completely missing from the main text (although I found it in Appendix). This ambiguity makes it impossible to determine the source of performance differences in Table 2. I cannot judge whether the observed differences are affected from the strong LLM models(compared with poor LLMs) or the architecture or the dataset. It needs ablation study.
- Poor written. The appendix is overly redundant and cumbersome, which impairs readability and makes it difficult for reviewers to quickly locate key supplementary information. Moreover, the paper contains two separate sections on “Limitations,” which is structurally disorganized. This duplication or misplacement of content creates confusion.
- The paper mentions that the mask encoder approach weakens the ability of positional encoding. Specifically, the VAR[1] work from last year has proposed a training strategy for positional encoding that is highly relevant to this limitation. Maybe it is a good way to handle or projection the position embedding in images.

references:
[1] Visual Autoregressive Modeling: Scalable Image Generation via Next-Scale Prediction

**Questions:**

- See Weaknesses

---

### Official Review · Reviewer_M46e · 2025-10-31

**Soundness:** 3
**Presentation:** 3
**Contribution:** 3
**Rating:** 6
**Confidence:** 4

**Summary:**

This paper introduces URECA, a region-level captioning dataset explicitly designed to enforce uniqueness of captions at multiple granularities (objects, parts, backgrounds), and a companion URECA model that generates unique region captions via a decoupled mask–image processing strategy with dynamic mask modeling. The dataset is built by a four-stage automated pipeline: constructing a mask tree by IoU; top-down short captioning conditioned on parent context; bottom-up refinement that aggregates child details into parent captions; and uniqueness refinement that contrasts similar regions using DINOv2 features. The test set is additionally verified with an LLM judge (GPT-4o-mini). The model prepends learned mask tokens (from a lightweight mask encoder) to frozen image tokens, preserving global context, and uses adaptive tiling to avoid losing fine detail at varying scales. Experiments report SOTA on the proposed URECA benchmark and ablations on mask tokens, model size, and components.

**Strengths:**

The paper addresses a clear, under-served failure mode in region captioning: captions that could equally refer to multiple regions in the same image. It formalizes this need for intra-image uniqueness and proposes both a dataset and a model that are purpose-built for it. The pipeline is thoughtfully designed to inject context (top-down) and detail (bottom-up), then explicitly disambiguate near-duplicates (uniqueness refinement) — a plausible route to one-to-one region–caption mappings.
Empirically, the model improves over strong region-captioning baselines (e.g., ViP-LLaVA, Osprey, KOSMOS-2) on URECA across BLEU/ROUGE/METEOR/BERTScore and a semantic LLM-based metric (CLAIR). Qualitative results demonstrate both fine part descriptions and within-image disambiguation. Ablations isolate the contributions of the mask encoder and dynamic mask tiling; token-length and model size trends are also sensible.

**Weaknesses:**

1. Evaluation scope is narrow (home-field advantage). Most gains are reported on the authors’ dataset. It’s unclear how well the method enforces uniqueness on external region captioning sets or referring tasks when uniqueness is not explicitly baked into the supervision. Cross-dataset generalization is not reported in detail in the current excerpts.

2. LLM-as-judge and test-set construction: The test set verification uses GPT-4o-mini. This may bias evaluation toward models and texts shaped by similar distributions, and it risks circularity (LLM-generated/verified data evaluated by LLM metrics). Human studies or task-grounded disambiguation tests (retrieval or localization via caption) would strengthen claims.

3. Uniqueness definition and measurement: While the paper defines uniqueness conceptually, a formal operationalization beyond LLM metrics and n-gram overlap is not fully specified in the excerpts. For instance, a caption could be “unique” yet still allow ambiguity if another region partially satisfies discriminative attributes. A more grounded metric (e.g., retrieval of the correct mask among hard negatives using the generated caption) would be compelling.

**Questions:**

1. Dataset statistics & splits. Please report per-split image counts, average masks per image, average levels in the mask tree, and the proportion of parts/background regions. Also share domain sources and any filtering for duplicates. This is necessary to assess coverage and bias.

2. LLM-judge robustness. Did you test with multiple judges (e.g., different families, temperature seeds) and measure agreement? Any human spot-checks to calibrate over-/under-penalization for “near-unique” captions?

3. Failure analysis. What are the dominant failure modes (e.g., tiny/thin structures, crowded scenes)? Please include an error taxonomy with representative images and captions.

---

### Note · Authors · 2025-11-27

I have read and agree with the venue's withdrawal policy on behalf of myself and my co-authors.